# Uncertainty Detection in Supervisor–Operator Audio Records of Real Electrical Network Operations

Jaime Acevedo [1], Gonzalo Garcia [1], Ricardo Ramirez [1], Ernesto Fabregas [2], Gabriel Hermosilla [1], Sebastián Dormido-Canto [2,*] and Gonzalo Farias [1]

1   Escuela de Ingeniería Eléctrica, Pontificia Universidad Católica de Valparaíso, Av. Brasil 2147, Valparaíso 2362804, Chile; jaime.acevedo.salinas@gmail.com (J.A.); gonzalo.garcia@pucv.cl (G.G.); gabriel.hermosilla@pucv.cl (G.H.); gonzalo.farias@pucv.cl (G.F.)
2   Departamento de Informática y Automática, Universidad Nacional de Educación a Distancia (UNED), Juan del Rosal, 16, 28040 Madrid, Spain; efabregas@dia.uned.es
*   Correspondence: sebas@dia.uned.es

**Abstract:** The quality of verbal communication, understood as the absence of uncertainty in the message transmitted, is a key factor in mission-critical processes. Several processes are handled by direct voice communication between these endpoints and any miscommunication could have an impact in success of the task. For that reason, the quality control of verbal communication is required to ensure that the instructions issued are effectively understood and adequately executed. In this context, it is expected that instructions from the command center are issued once, and that the acknowledgment from the field are minimal. In the present work, the communication between an electrical company control center and factory workers in the field was chosen for analysis. We developed two complementary approaches by using machine learning and deep learning algorithms to assess, in an automatic way, the quality of information transmission in the voice communications. Preliminary results demonstrate that the automatic uncertainty detection is feasible, despite the small number of samples available at the present time. To support further studies, a repository was created in GitHub with the spectrogram and the tokenized words of all audios.

**Keywords:** uncertainty; spectral analysis; speech analysis; support vector machine (SVM)

## 1. Introduction

The management of electrical networks is a mission-critical process [1,2], and for this reason, efforts must be continually made to eliminate sources or inducers of error, particularly in the communication between the control center and the operating personnel on the ground [3]. In general, the communication goes from the command center to the field worker. Instructions are given to the deployed personnel, and they normally acknowledge correct reception or understanding. Problems in this communication normally imply either delays in the fixing of some failed device, restoring services, or in more extreme cases, a fallen service for unnecessary extra time. In the field of electrical network management, a control center is an organizational unit responsible for the supervision and coordination of the real-time operation of the electrical system. Its function is to ensure the continuity of supply, and the security of that system, its assets and the people involved in it, and keeping the technical parameters within the defined legal ranges, as can be seen in [4].

To fulfill the objective of the control center, there is the role of operator, which is the person in charge of monitoring and remotely commanding the electrical system through field equipment, telecommunications, and SCADA [5]. In case there is the need to carry out face-to-face work in the field, they are responsible for remotely instructing the necessary maneuvers to carry out the work safely.

For these complex systems, where there is a close interplay between machines and human beings, sources of errors are not only circumscribed to the automatized processes

but often arise in the human-to-human level interaction (see ([6,7]) for communication problems in electricity company management). In daily operations, voice communication is common between a control center and personnel deployed to work on distributed elements throughout the city. These communications are normally recorded for future reference, if needed, and normally do not serve as an active tool for performance or service quality robustness assessment. Close attention in the post-analysis of these audios, especially after the occurrence of a problem/failure, tends to show a correlation between the problem and some level of hesitancy or uncertainty in the communications in relation to it. This is true in spite of the absence of clear indicators of doubt on any side of the line, where the conversations could be evaluated as successful by a third party based on the content of the speeches.

It is for this reason that there are strict communication protocols between the operator and the field personnel so that the instructions issued are effectively understood and adequately executed. Despite the existence of these definitions, the complexity of verbal communication generates certain risks when there is not good understanding between the interlocutors. This complexity is given, among others, by the work to be carried out, the moods, the context, and the technical quality of the communication channel and its terminal equipment. To mitigate the risk of errors in the execution of the work, training is carried out in the aim of ensuring effective communication.

To evaluate the results in practice, random reviews are made of the recordings of these verbal interactions. This method, while effective for individual-reviewed recordings, has the following disadvantages:

- High cost in time, since listening to the recordings takes at least the same time as the original communications.
- Low coverage, since it is only possible to review a smaller subset of the recordings, thus losing the opportunity to capture a greater number of situations to improve.

In order to overcome the disadvantages of the current method, and thus increase the effectiveness of the quality control process within the company's electrical work management, the use of an automatic recording analysis process is proposed to:

- Review the totality of the recordings, regularly obtained at a rate of over 5000 per month, with an average of 130 s per audio. These statistics are expected to continue to steadily increase with time.
- Classify communications that generate a risk of error due to lack of clarity. These errors are normally translated into the need for repeated work in the field, preventable machine failures, or hazardous conditions for personnel.

The detection of these hidden uncertainty inducers is then a good indication of a potential problem if obtained opportunely. The proposed idea in this article is the extraction of useful data from both the spectrum content and from the composition of the words within the text of the audio files to allow an inference of the existence of hidden uncertainty.

In this work, two complementary methods are detailed. The first one, labeled formant analysis, is based on extracting the relevant frequency content per sample time, the first two formants, from the spectrogram, and determining the degree of uncertainty based on how these formants change throughout time (see [8,9] for related works with audio frequency contents). The second method called speech-to-text, is basically inferring the existence of uncertainty in the order of words used in the audio, reflecting the sentiment of the speakers, as can be seen in [10] for a close related approach.

In the first approach, the design of an uncertainty classifier is proposed, based on the identifications of phonemes present in situations of uncertainty characterized by an extended repetition of vocals or consonants, such as 'eeeeeeee...', or 'mmmmmmm...'. These situations are identified through the analysis of the formants, which correspond to the frequency of the largest magnitudes for each time instant of the communication [11–13]. Speech phonemes are described by the first and second formants, which are obtained by analyzing the spectrogram of the audio file containing the communication records. Previous

works have used deep learning and linear predictive approaches estimating frequency contents in speech processing, as can be seen in [14,15]. Unchanged or slightly changed formants between consecutive time instants are indicative of unaltered frequency contents, which in turn are a consequence of fixed sonarization.

The second approach, primarily based on detecting the hidden "sentiment" or "emotion" content, in this case, associated with hesitation or doubts, uses the order in which the words are included in the text of the audio. Related works include [16,17]. This approach is based on a technique called speech-to-text [18,19]. In this case, it is important to consider that a process trained for the Spanish language should be used, ideally Latin American or Chilean, so a special opener tool was used to carry out this audio-to-text conversion. A bag of words was then constructed from all words existing within the audio, and consequently, a process of converting the words into numbers, called tokenization, based on a Tensorflow API, was performed for effective numerical analysis. Similar performance metrics are employed in this second method.

Authors have applied these approaches in previous works, showing a remarkable capacity in terms of sensible classifications ([20,21]). A common pre-processing step is the obtaining of what is called vector of characteristics (VC) from the raw data, which in this case, captures the audio's main features while reducing their high dimensionality, which is then used to train support vector machines (SVMs) and deep neural networks (DNNs). Through a process based on the analysis of the formants, the first approach generates the characteristic vectors (see [22]), while the second approach base its vectors on the text content, and then the SVM and DNN algorithms (see [23–25]) are used to obtain a classifier for uncertainties. Several experiments, with different parameters, have been performed in order to measure the effectiveness of the SVM through performance metrics, as can be seen in [26]. Finally, the parameters that generate the best SVM and DNN performance for the case under study, are identified. This allows the fulfillment of the objective of detecting communications that have levels of uncertainty that require review by a supervisor within a context of the continuous improvement of the overall process.

The implementation is based on machine learning and deep learning algorithms executed in batch mode (offline), processing audio files with recordings of the dialogues with instructions. By extracting features and using both approaches, the quality of the interaction between the control center and the field operator will be classified. The proposed idea is expected to be implemented in real-time in the near future. The main contributions of this paper can be summarized as follows:

- Two different approaches, based on pattern recognition, are used to classify the audio, including a technique called support vector machine, and deep neural networks. Both approaches use different processing techniques, mainly based on frequency content analysis, text composition, and natural language study.
- The use of these machine learning techniques allowed the successful classification and differentiation of audios with uncertainty from a set of audios labeled audios.
- As a way of supporting the analysis of audios under possible uncertainties, the audios used in this research, including their spectrograms, and text tokenization, are published in GitHub ([27]), and remain accessible for further work.

The rest of the paper is structured as follows. Section 2 introduces the problem statement, conceptualizing the problem to be solved and its relevance. Sections 3 and 4 present the two methods covered in the article and their technical details. And, Section 5 presents the results obtained from training, testing and validating both methods and their complementary natures. Section 6 provides the conclusions.

## 2. Problem Statement

The real-time operation of an electrical system qualifies within critical mission processes. For this reason, any error or misunderstanding can generate serious undesired consequences, both in the continuity and quality of the supply, as well as in the state of the assets, the health of the workers, and finally, have an impact on the population and

the reputation of the company. That is why the inducers of undesired events must be minimized, with verbal communication between the operator and field personnel being one of the most sensitive aspects, given the direct interaction between two people, as can be seen in Figure 1.

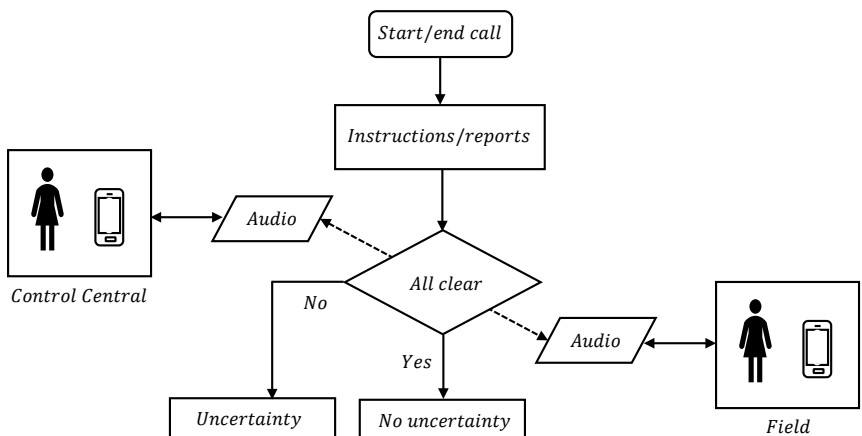

**Figure 1.** Communication between central control (supervisor) and field (operator).

From a preventive and anticipatory perspective, it is of interest to identify real communications that generate the risk of error in the execution of manual operations in the field in order to carry out the interventions and feedback that allow eliminating these risks of misunderstandings or confusing or erroneous instructions. The direct benefit translates into a reduction in time and an increase in the efficiency and effectiveness of the review of audio files since the designed classifier must eliminate those conversations without uncertainty or with a low probability of uncertainty, selecting and sorting those audios that have greater detected uncertainty.

The objective of this work is to identify the audio recordings that are most likely to contain communications with uncertainty. Therefore, the problem to be solved is stated as how to identify the uncertainty of a verbal communication recorded in an audio file, so then tackled by the two proposed complementary approaches:

- *Formant analysis*: The implementation of the first method is carried out through a complex process based on the identification of phonemes and their formants that represent uncertainty in communication, for example, 'huuuuuh', 'eeeeeeh', 'mmmmmmm'. Formants correspond to the strongest frequencies present in an audio signal. Formants are obtained by means of analysis in the frequency domain through a spectrogram that represents the intensity of the signal for each frequency within its bandwidth and for each discrete time instant of the recorded signal. In Spanish, the phonemes are determined by the frequency and magnitude of the first two formants, as can be seen in [12,28]. Thus, there are the so-called letters of formants (LoF), which allow us to identify the use of vowels. See Figure 2 for the LoF for vowels in Spanish.
- *Speech-to-text analysis*: The implementation of the second method is performed through a speech-to-text process. The identification of words or sets of words that represent uncertainty in communication, by defining a bag of words and their order within, that should not be used or repeated in the context of critical mission instructions. For example: 'No', 'I do not know', 'Wait', 'I'm going to find out', 'I do not understand you', 'Are you sure?' or others. See Table 1 with sentence samples of some audios labeled as "with uncertainty" and "without uncertainty" by an expert, signaling on one hand a level of hesitancy and vacillation, or clear certainty on the other. In this case, it is important to consider that a process trained for the Spanish language should be used, ideally Latin American or Chilean. Once the written text has been obtained, the use of words or phrases contained in a list of words that generate uncertainty is identified.

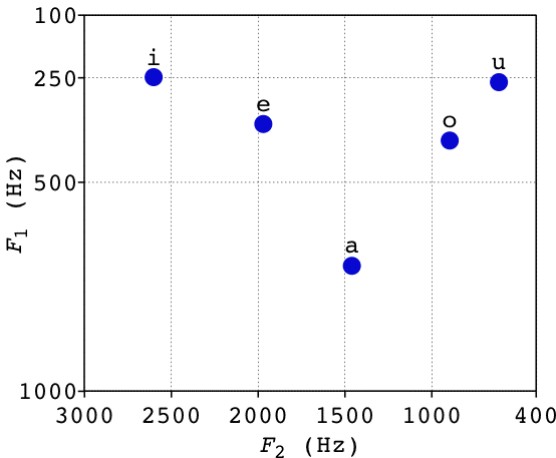

**Figure 2.** LoF for vowels in Spanish [12].

**Table 1.** Sample sentences from audios that do or do not reflect the existence of uncertainty.

| Audio | Sentence (Spanish/English) | Uncertainty |
|:---:|:---:|:---:|
| 5 | "…ah, listo, te entiendo, te entiendo…"<br>"…oh, done, I understand you, I understand you…" | no |
| 62 | "…correcto, sí, correcto, ya…"<br>"…correct, yes, correct, OK…" | no |
| 73 | "…ya, ah ya, perfecto…"<br>"…OK, oh OK, perfect…" | no |
| 13 | "…no te aseguro…"<br>"…I do not assure you…" | yes |
| 22 | "…no sé, OK, vamos a averiguar…"<br>"…I do not know, OK, we'll find out…" | yes |
| 28 | "…a ver, esperame…"<br>"…let us see, wait for me…" | yes |

During training, both methods considered in this work were evaluated through the following machine learning metrics for classification problems (see [29] for similar application). These metrics are based on $TP$, true positives, namely audios tagged with uncertainty and correctly classified; $TN$, true negatives, audios tagged without uncertainty and correctly classified; $FP$, false positives, audios tagged with uncertainty but incorrectly classified; and $FN$, false negatives, audios without uncertainty and incorrectly classified:

- *Precision*: The ratio of audios correctly classified with uncertainty over all audios was classified with uncertainty. A precision of 100% means that all audios classified as "with uncertainty" are actually "with uncertainty". Precision is also known as reliability.

$$Precision = \frac{TP}{TP + FP} \tag{1}$$

- *Recall*: The ratio of audios correctly classified with uncertainty over all audios with uncertainty. This is also known as the true positive rate (TPR). A recall of 100% means that all audios "with uncertainty" are correctly classified as "with uncertainty".

$$Recall = \frac{TP}{TP + FN} \tag{2}$$

- *Accuracy*: The ratio of audios correctly classified over all existing audios.

$$Accuracy = \frac{TP + TN}{TP + FP + TN + FN} \tag{3}$$

- $F_1$ score: The harmonic average of precision and recall.

$$F_1 = 2\frac{Precision \times Recall}{Precision + Recall} \tag{4}$$

## 3. Formant Analysis Approach

Although the use of the LoF allows us to identify the use of vowels in Spanish, in this case, it does not consider the use of consonants that are usually used to cover up uncertainty. For example, it is using 'mmmmmm' while searching for an answer.

Given the above disadvantage, rather than searching for vowels using the Formant Chart, we will seek to identify audio sections that have similar contiguous formants since this represents the use of the same phoneme. This strategy is based on the fact that it is not of great interest to identify which phoneme is used but rather to identify a phoneme used during a certain window of time. Once exceeded in duration, a sign of uncertainty in verbal communication can be assumed. The implementation process of this strategy, depicted in the block diagram of Figure 3, is as follows:

- Extract the data from the audio file consisting of the digital samples and the sampling frequency.
- Calculate the spectrogram of the sample set, where design parameters could be the size and shape of the moving window for the discrete Fourier transforms.
- For each time instant, extract the first and second formants of the respective squared spectrum. These correspond to the two highest amplitudes, indicating the frequencies and amplitudes.
- A threshold noise comparison is performed to eliminate formants with amplitudes low enough that could be corrupted by noise.
- The level of uncertainty is extracted by adding up the time intervals where the frequencies of the filtered formants do not change, creating three categories with varying degrees of uncertainty.
- For each file, a vector of characteristics (VC) is put together, including the three levels of uncertainties and the fraction they cover over the entire signal.
- The VCs are used to train an SVM, considering that audio files are previously manually evaluated and tagged as with or without uncertainty.

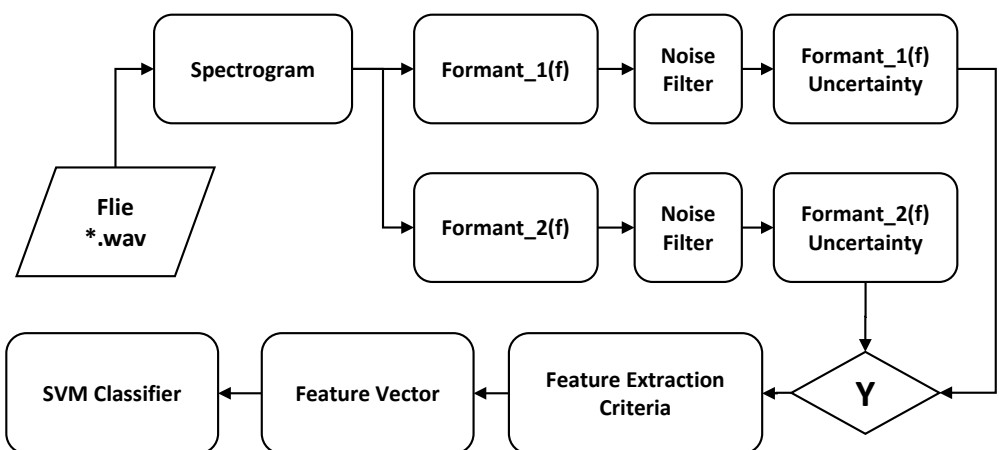

**Figure 3.** Formant analysis uncertainty classifier process.

### 3.1. VC Creation

From the formant detection process, the following formant matrix, $M_F$, is extracted:

$$M_F = \begin{bmatrix} 1 & f_1^1 & m_1^1 & f_1^2 & m_1^2 \\ 2 & f_2^1 & m_2^1 & f_2^2 & m_2^2 \\ \vdots & \vdots & \vdots & \vdots \\ N & f_N^1 & m_N^1 & f_N^2 & m_N^2 \end{bmatrix} \tag{5}$$

with the first column containing the sample index ($N$ samples), the second and third columns containing the frequency and normalized magnitude of the first formant, respectively, and the fourth and fifth containing those related to the second formant. As explained, it is interesting to detect audio sections where the phoneme remains nearly constant. This means that audio windows meet the following condition for $i = 1, 2$

$$|f_{n+1}^i - f_n^i| < f_{max} \tag{6}$$

which is associated with a level of uncertainty. The parameter $f_{max}$ is the maximum frequency deviation, a tolerance parameter used to detect very similar frequencies, but not necessarily identical, given the slight tonal variations of the voice for the same syllable. The formant's magnitude is not considered to affect the analysis, given that its value is not below a noise threshold.

By using a reference audio file, which contains the five vowels, it is empirically determined that a suitable value that allows one to distinguish similar contiguous phonemes is $f_{max} = 50$ Hz. Smaller values generate a model with low sensitivity, while larger values tend to classify unrelated phonemes as uncertain. The 50 Hz chosen in this study corresponds to the voice of a male adult. But this value should vary according to the gender, age, and the characteristics of each person's voice.

The advantage of the calculation of a spectrogram over a regular spectrum of the entire voice signal, is its time–frequency nature, especially adequate for non-stationary signals. A series of spectra are obtained by calculating the short-time Fourier transforms for short overlapping time intervals of the signal. This captures how the frequency content changes with time, as opposed to a single spectrum for the entire signal, where there is no easy way to differentiate any time change. As an illustration of the underlying assumption, the spectrogram of an audio file is calculated, as shown in Figure 4. The first and second formants are shown in Figure 5.

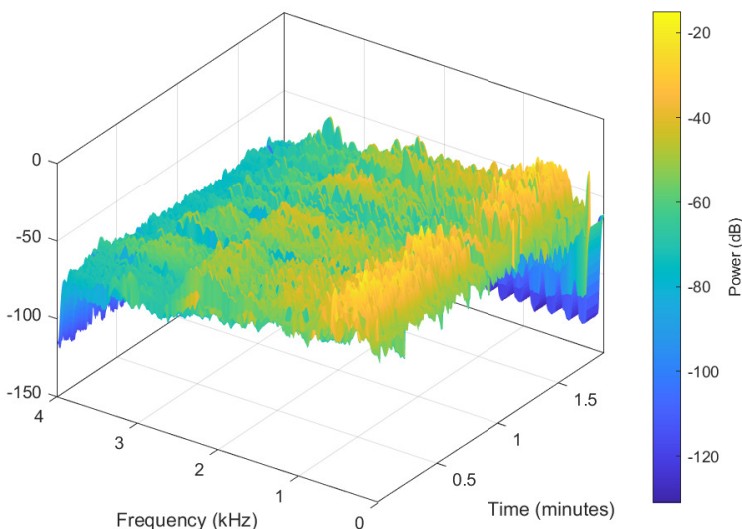

**Figure 4.** Spectrogram of an audio file.

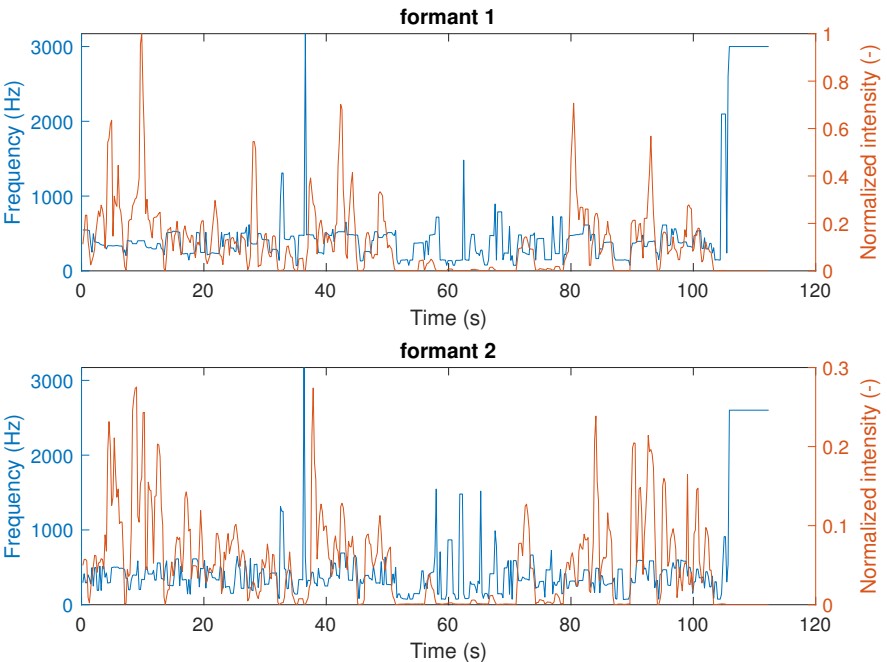

**Figure 5.** First and second formants of an audio file.

The formants are filtered by comparing their intensity to a threshold, in this case, 10%, discarding everything below it. These parameters were tuned by experimentation. Figure 6 shows a section of the filtered formants, between the fourth and seventh seconds, where both formants have a nearly constant frequency (with deviations less than $f_{max}$), which is reflective of a fixed phoneme and an associated level of uncertainty, considering a minimum level of their magnitudes. The algorithm then extracts the time length of the sections of the audio with this condition and generates the vector of characteristics.

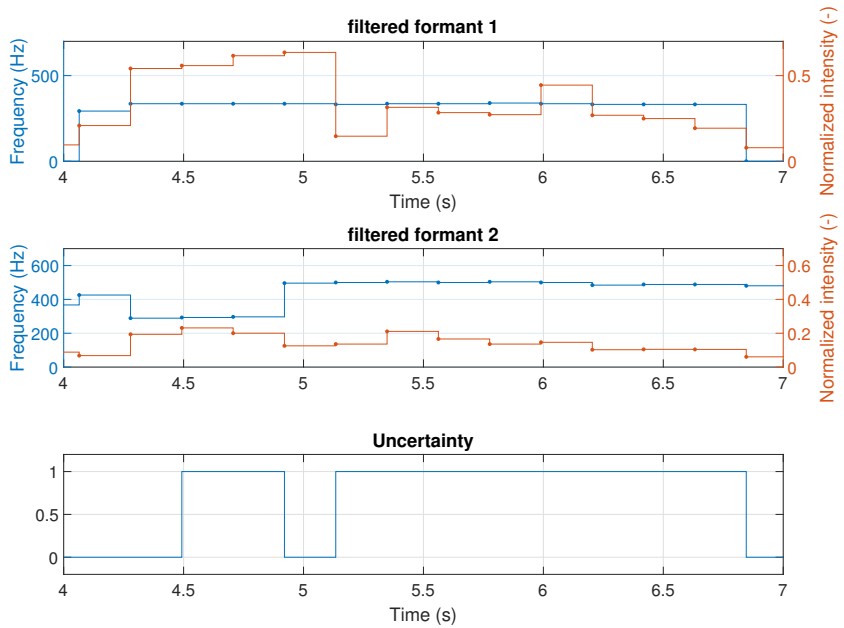

**Figure 6.** Filtered formants showing the possible existence of uncertainty.

This vector is defined by the total duration of the sections of similar phonemes, as an uncertainty interpretation, using the following criteria:

- $T_1$: cumulative time of similar filtered frequencies whose uninterrupted duration is contained between 200 and 300 ms.
- $T_2$: cumulative time of similar filtered frequencies whose uninterrupted duration is contained between 300 and 500 ms.
- $T_3$: cumulative time of similar filtered frequencies whose duration is greater than 500 ms.
- $\delta_T$: percentage of time with uncertainty $(T1 + T2 + T3)$ with respect to the total duration of the audio $T$.

The VC is then defined by

$$VC = \begin{bmatrix} T_1 \\ T_2 \\ T_3 \\ \%_T \end{bmatrix} = \begin{bmatrix} \sum_{n=1}^{n_{T_1}} t_1^n \\ \sum_{n=1}^{n_{T_2}} t_2^n \\ \sum_{n=1}^{n_{T_3}} t_3^n \\ \frac{(T_1 + T_2 + T_3)}{T} \end{bmatrix} \tag{7}$$

with $n_{T_i}$ the number of occurrences of $T_i$, and $t_i^n$ the uninterrupted time of each occurrence, for $i = 1, 2, 3$. The VC extracted for the current audio, with $T = 112.52$ s, was $VC = \begin{bmatrix} 4.92 & 7.27 & 13.26 & 22.62 \end{bmatrix}^T$.

### 3.2. SVM Training

To perform the classification of audio into "with uncertainty" and "without uncertainty", the supervised machine learning approach support vector machine (SVM) is selected. The machine is fed a set of training VCs, and their corresponding binary labels (with or without uncertainty), previously determined. During the training, a model is generated which, after completion, is used to classify unlabeled new videos. This mapping creates a hyperplane that maximizes the separation of the two categories.

Given a normalized training set of $m$ points

$$(\overline{VC}^1, y^1), \ldots, (\overline{VC}^m, y^m) \tag{8}$$

with $y^i = \{-1, 1\}$, for no uncertainty and uncertainty, respectively, the tag associated to $VC^i$ and audio $i$, the training step delivers the hyperplanes splitting the entire vector space into two sets: the ones either above the higher hyperplane being classified as with no uncertainty, or below the lower hyperplane, classified as with uncertainty. Another important information delivered by this method, although not used in the present work, is not only the segmentation of the data points into one or the other condition, but also the distance from the hyperplanes can give some insights into the degree of membership. For the training, a linear Kernel function was used, and a cost of penalty factor (box constraint).

## 4. Speech-to-Text Analysis Approach

The use of certain words, their frequencies, and their location within the audio are expected to carry or reflect some degree of uncertainty. Converting these words into numbers is then used to train a deep neural network as well as an SVM.

### 4.1. Analysis Based on DNN

This method is based on the training of a multilayer perceptron (MPL) deep neural network to classify the uncertainty. The preprocessing of the raw data is different from the first method. Here, the audio is converted into text by a process called speech-to-text. Whisper, developed by openAI [30], was used to carry out this audio-to-text transformation process. This system is trained with a variety of languages, one of which is Spanish, which is the one used, given the audio Spanish languages, achieving a very high rate of transcription success. Once the written text has been obtained, a list of words is built with an identification number associated with each one. This process, called a tokenizer, is explained below.

The resulting numeric vectors, each one associated with a particular audio file, are ready to be fed into the DNN for training. As in the previous method, the tagged information, with or without uncertainty, is also used as the target. Figure 7 shows a block diagram with the basic steps during training.

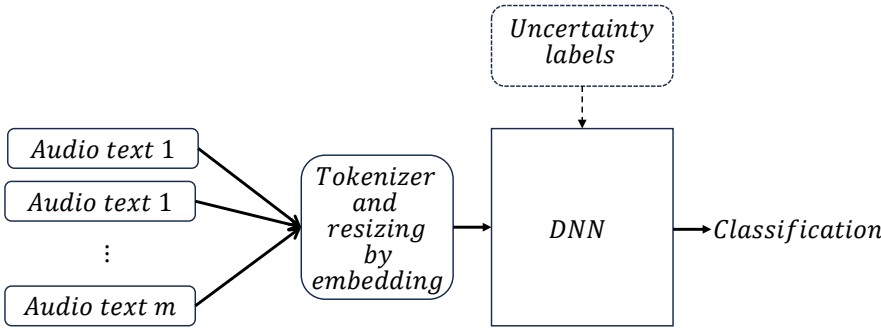

**Figure 7.** Speech-to-text analysis uncertainty DNN classifier process.

The network is built using the Keras Sequential model [31], due to its simplicity in building it by stacking different types of layers, and for the high dimension of the input data.

### 4.1.1. Tokenizer

This process, based on a subfield of linguistics called natural language processing, consists of converting words into a corresponding unique number as an identifier. A bag of words is built by assigning these numbers to each word without repeating them. Once the dictionary is built, the texts are just converted into numeric vectors $VN_{DNN}^i$, $i = \{1 \ldots m\}$, as sketched in Figure 8. This was achieved using Tensorflow API [32]. These vectors will be further reduced in dimensionality by the embedding stage, at the entrance of the DNN.

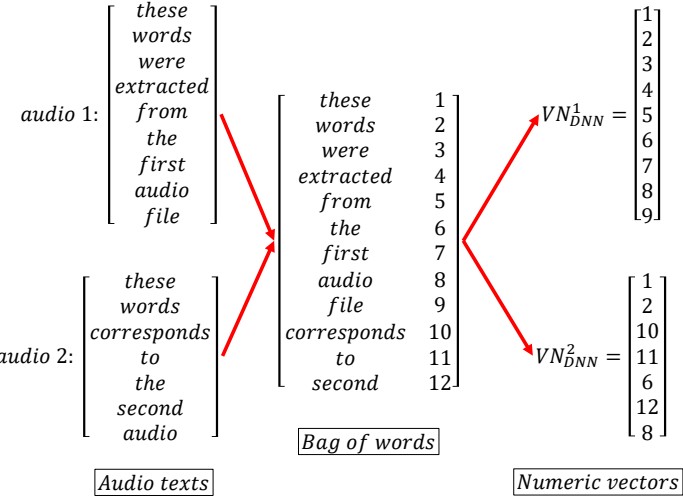

**Figure 8.** Tokenizer process for two arbitrary texts (original audios were processed in Spanish).

Two main concerns are given as follows. Due to the variable nature of the data, all the vectors are resized by adding null numbers (zeros in this case), and matching their sizes, called zero padding. The second concern is that the corresponding bag of words to be adjustable by incorporating new words was not seen during the training.

### 4.1.2. DNN Training

This part of the process corresponds to standard DNN training, where its weights and biases are numerically searched in an iterative way, by back-propagating the estimation error.

The DNN model is based on a Keras Sequential model as shown in Figure 9. Its layers, number of neurons each, activation functions, number of epochs, and embedding dimensions were obtained by fine-tuning and optimizing its inference power. Together with the dense layers, dropout and regularizer L2 layers were added to its high sensitivity to balance within the data, avoiding over-fitting. For training, the following hyper-parameters were used: at the input layer, an embedding dimension of 150; at both dense layers, 60 neurons, and regularizers of 0.01; and dropout factors of 0.2. The total number of epochs was 150.

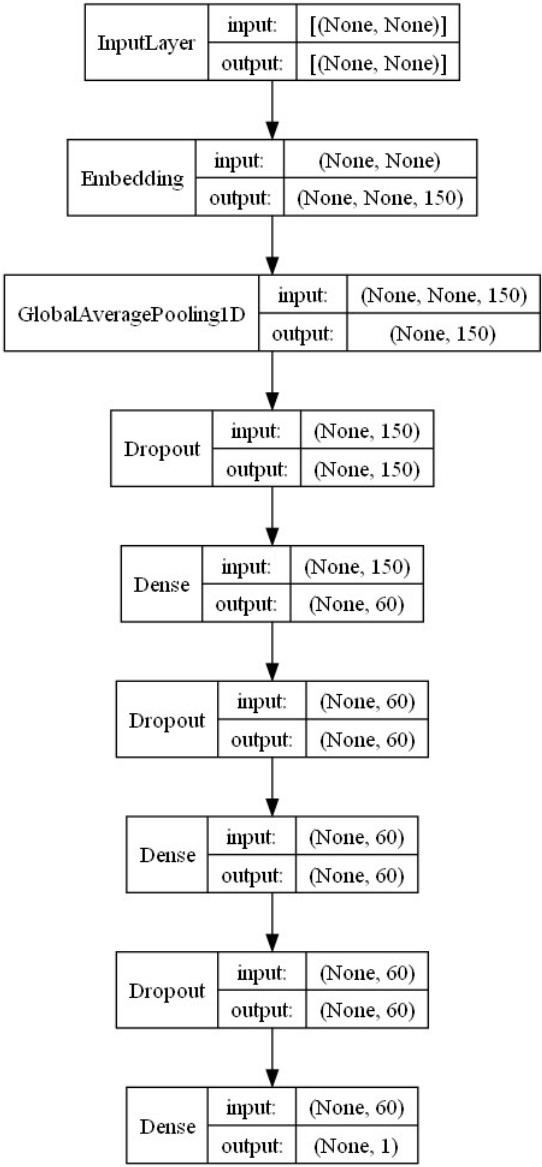

**Figure 9.** Keras sequential model.

### 4.2. Analysis Based on SVM

Data gathered from the speech-to-text approach were also used with an SVM. In this case, the characteristic vectors to be injected into the SVM classifier include the relevant repetition frequency of each word within the totality of the audio, frequencies that have surpassed a tuned threshold. First filtering is the discarding of words from the bag of words, with three letters or less, in an attempt to clean the data from words with functions more related to connecting ideas or sentences, like most prepositions. Further looking to concentrate the most important features contained within the repetition frequencies,

and reducing the overall data dimensions, these numeric vectors are subjected to a technique called non-negative matrix factorization [33]. These new numeric vectors, $VN_{SVM}^{i}$, $i = \{1 \ldots m\}$, are then fed into the SVM engine. This is sketched in Figure 10. Similarly, as in Section 3.2, for the training, a linear Kernel function was employed, with a cost of penalty factor.

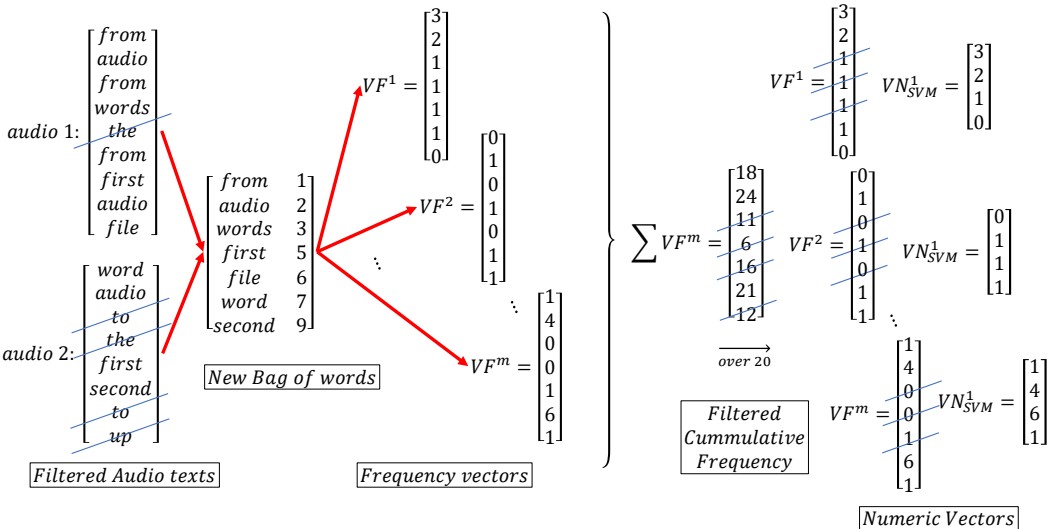

**Figure 10.** Frequency vector creation from two arbitrary texts (original audios were processed in Spanish).

### 4.2.1. Data Filtering by Relevant Repetition Frequency

Here, the already computed bag of words as well as the tokenization for the ANN are used. The difference is that, instead of resizing them by the embedding method, the data are mapped into their repetition frequencies. The cumulative repetition frequency per word within all audios is calculated. A threshold is then defined to select those frequencies above it, and their associated words potentially carry an inherent sentiment to be associated with the level of uncertainty.

### 4.2.2. Non-Negative Matrix Factorization and K-Fold Cross Validation

This technique, also called non-negative matrix approximation (NNMF) (see [34] for more details), performs linear algebra multivariate analysis, wherein a matrix is factorized into two matrices, all elements begin as non-negative and undergo reducing dimensionality, which helps one work with the high volumes of data.

The data are randomly partitioned into equal sized subsets. Out of the subsets, a unique subset is kept from the training process, but used later for testing. This process is repeated the same number of times as the partition, interchanging the subset chosen for testing, so that all are considered. The results are then averaged (see [35]).

## 5. Experimental Results

To train the algorithms, there is a set of 190 tagged audio recordings, including 31 labeled as "with uncertainty" and 159 labeled as "without uncertainty". With that universe of labeled recordings, and given the unbalanced number of audios with and without uncertainties, different proportions of files of each class are tested to assess the effect of this imbalance. These proportions are shown in Table 2.

To obtain statistically more representative results, 100 runs were performed for each of the combinations shown in Table 2, selecting the subset without uncertainty randomly out of the 159. The results were averaged per each category. Mean values are presented.

**Table 2.** Five different proportions of "with uncertainty" audios of the total dataset.

| With Uncertainty | Without Uncertainty | Percentage |
|---|---|---|
| 31 | 159 | ≈16% (31 out of 190) |
| 31 | 66 | ≈32% (31 out of 97) |
| 31 | 33 | ≈48% (31 out of 64) |
| 31 | 18 | ≈64% (31 out of 49) |
| 31 | 8 | ≈80% (31 out of 39) |

Given the imbalanced nature of the available data, different proportions of audio were tested in order to obtain some insight into the effect of this feature. In theory, and this was detected from the results, better results were obtained when the proportions were more equivalent. This was the case in both methods.

Tests were designed including all 31 audios with uncertainty, and a varying number of audios without uncertainty (see Table 2), corresponding to the percentages: 16%, 32%, 48%, 64%, and 80%, with the 16% corresponding to the case where all 159 audios without uncertainty were used. For each of these cases, up to 100 runs were executed by randomly shuffling the audios without uncertainty to be included. The resulting metrics, namely precision $P$, recall $R$, accuracy $A$, and $F_1$, per each case, were averaged.

For each case, validation was performed after training. Out of the total number of audios, this is the 31 with uncertainty plus the particular number used that were without uncertainty; a randomly selected partition was made to set aside a fraction for the actual training, and the remaining fraction for validation. This is, after finishing the training, the algorithm which was presented with new data, that were not used during the training but belonged to the current case.

For each of the cases shown in Table 2, and each of the 100 runs, the following partitions were set to divide audios for training and for validation: 10%, 20%, 30%, 40%, 50%, 60%, 70%, 80%, and 90%, where the percentage indicates the number of audios selected for validation out of the total.

### 5.1. Formant Analysis Results

According to the runs resulting from the variation of previously indicated parameters, the performance curves of the precision, recall, accuracy, and $F_1$ for the formant analysis method are shown in Figure 11. From the analysis of the performance curves of all metrics, the following can be observed:

- For the cases with low proportions of "with uncertainty" (16% and 32%), the performance is in general very low.
- For the cases with intermediate proportions (48% and 64%), the performances are better and more stable for different validation ranges in comparison with the other ones.
- The proportion with 80% seems to be the best result for the formant analysis method. Table 3 details the metric performance for 80%.

**Table 3.** Formant analysis metrics for 80% proportion (see Table 2).

| Validation (%) | Precision | Recall | Accuracy | $F_1$ |
|---|---|---|---|---|
| 10 | 1.0000 | 0.9300 | 0.9300 | 0.9550 |
| 20 | 0.8680 | 0.9083 | 0.7986 | 0.8812 |
| 30 | 0.8276 | 0.9189 | 0.7745 | 0.8672 |
| 40 | 0.8153 | 0.9017 | 0.7553 | 0.8521 |
| 50 | 0.8154 | 0.9060 | 0.7611 | 0.8547 |
| 60 | 0.8333 | 0.8641 | 0.7474 | 0.8430 |
| 70 | 0.8300 | 0.8666 | 0.7496 | 0.8430 |
| 80 | 0.8181 | 0.8233 | 0.7126 | 0.8134 |
| 90 | 0.8173 | 0.4229 | 0.4766 | 0.7820 |

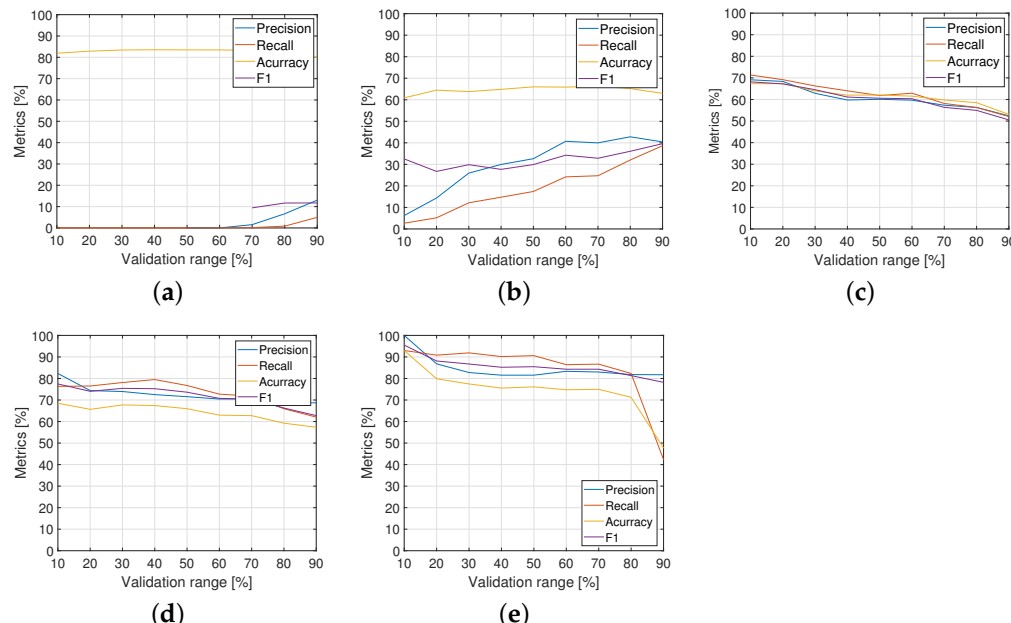

**Figure 11.** Formant analysis metrics for proportions shown in Table 2: (**a**) 16%; (**b**) 32%; (**c**) 48%; (**d**) 64%; (**e**) 80%.

### 5.2. Speech-to-Text Results

The results of the application of SVM and DNN techniques, described below, can be seen as complementary.

### 5.2.1. Based on DNN

The results of this method for the same metrics precision, recall, accuracy, and $F_1$ are shown in Figure 12. The following can be deduced:

- When there is an imbalance in the data in the training sets (16% and 32% of data 'with uncertainty'), good results are not obtained, since there is an over-fitting of the model to the majority group, in this case corresponding to the data 'without uncertainty'.
- For balanced training sets (48% and 60% of data 'with uncertainty'), significantly higher results are achieved, with 90% and 98%, respectively. Still, it is impossible because the test set is so low (10% validation). This case is expected to be fully representative.
- When the validation percentage is between 20% and 40% in balanced training sets (48% and 60%), the best results are obtained, namely 97% and 88% in the best cases, respectively.
- For the percentages mentioned in the previous point, an average of all the iterations was not performed, but the results that obtained a high percentage of recall as well as a high percentage of total success were extracted since it is not appropriate to have a high percentage of recall and close to 50% in the model because it would be in front of a highly over-fitted model.
- Although a large percentage is achieved in the case of 60%, the hyperparameters of the neural network had to be constantly adjusted to avoid over-fitting the data.
- Models with a balanced training set suffer as the percentage of validation data increases since this means that fewer data are used for training, and in this context, one works with a limited dataset for the class 'with uncertainty', so the model is negatively affected the fewer data are used.
- The 80% proportion is having the optimal results for the speech-to-text DNN analysis method. Table 4 details the metric performance for this.

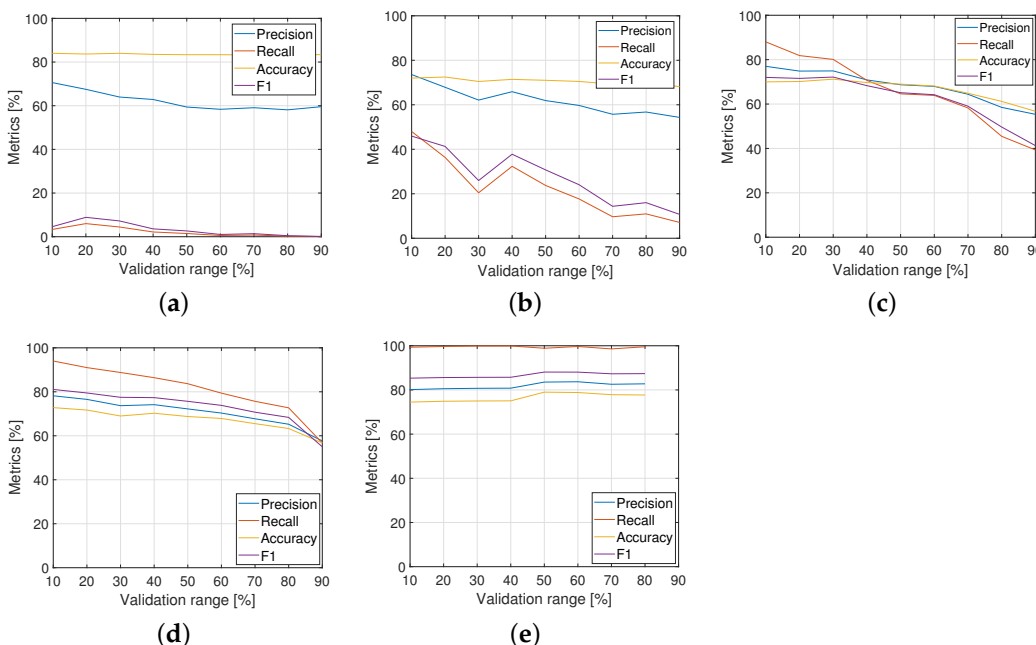

**Figure 12.** Speech-to-text DNN analysis metrics: (**a**) 16%; (**b**) 32%; (**c**) 48%; (**d**) 64%; and (**e**) 80%.

**Table 4.** DNN metrics for 80% proportion (see Table 2).

| *Validation* (%) | *Precision* | *Recall* | *Accuracy* | $F_1$ |
|---|---|---|---|---|
| 10 | 0.8014 | 0.9933 | 0.7450 | 0.8533 |
| 20 | 0.8055 | 0.9966 | 0.7487 | 0.8560 |
| 30 | 0.8073 | 0.9988 | 0.7500 | 0.8569 |
| 40 | 0.8079 | 0.9991 | 0.7506 | 0.8573 |
| 50 | 0.8355 | 0.9886 | 0.7900 | 0.8809 |
| 60 | 0.8369 | 0.9966 | 0.7882 | 0.8806 |
| 70 | 0.8257 | 0.9857 | 0.7785 | 0.8730 |
| 80 | 0.8276 | 0.9954 | 0.7770 | 0.8737 |

5.2.2. Based on SVM

This analysis is based on the repetition frequency of words in each text. Figure 13 shows the frequencies for both sets of audios together (from word 250 to word 450, out of the total of words out of 4000) associated with the subset of words in the bag of words. A threshold of 20 repetitions has been set to reduce the more relevant words based on their prevalence.

As before, data are analyzed based on Table 2, and then a non-negative matrix factorization (NNMF, [36]) is applied with 15 salient features. Each output is then 10-folded for classification. Similarly, the performance curves of precision, recall, accuracy, and $F_1$ for the speech-to-text DNN analysis method are shown in Figure 14. The following points can be stated:

- In a similar pattern, the performance is generally very low, for low proportions of "with uncertainty" audios (16% and 32%).
- The performance is relatively improved for intermediate cases (48% and 64%), as a function of validation ranges.
- As a confirmation of previous results, the 80% proportion gives better results overall. Table 5 details the metric performance for 80%.

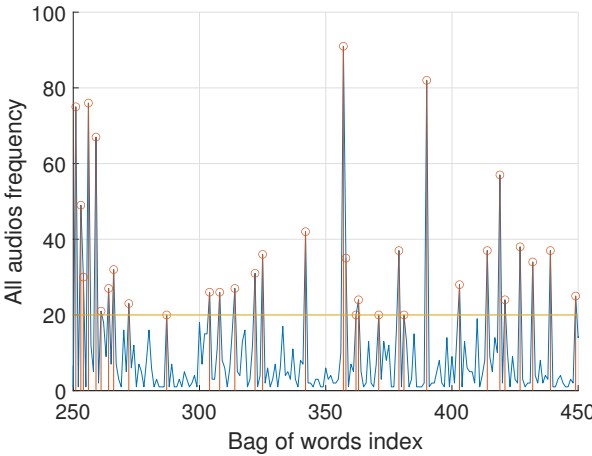

**Figure 13.** Repetition frequency of words in each audio text.

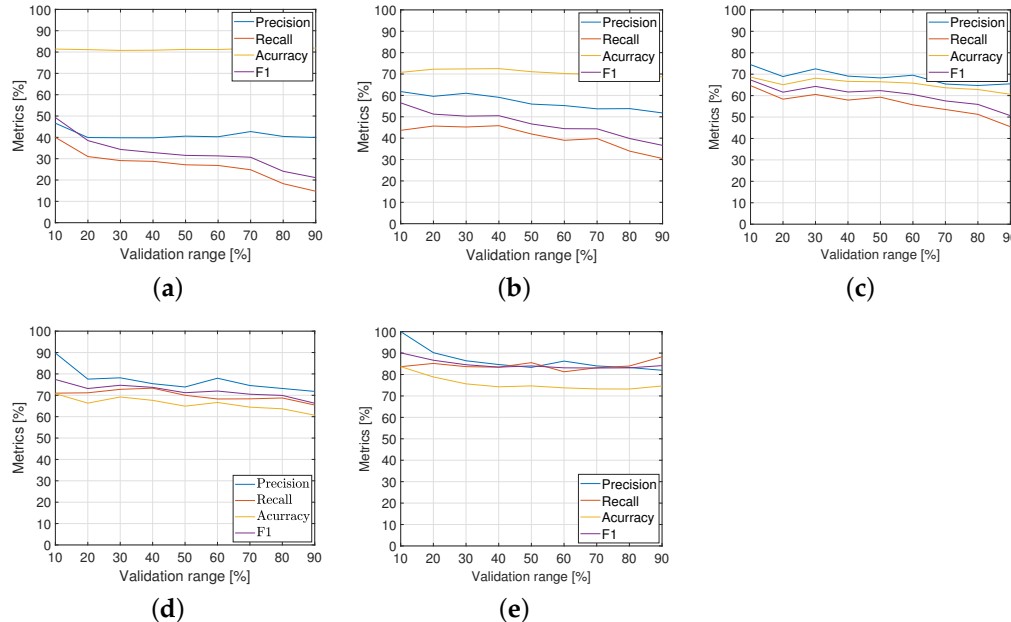

**Figure 14.** Speech-to-text SVM analysis metrics: (**a**) 16%; (**b**) 32%; (**c**) 48%; (**d**) 64%; and (**e**) 80%.

**Table 5.** SVM for frequencies, based on non-negative matrix factorization metrics for 80% proportion (as can be seen in Table 2).

| Validation (%) | Precision | Recall | Accuracy | $F_1$ |
|---|---|---|---|---|
| 10 | 1.0000 | 0.8366 | 0.8366 | 0.9010 |
| 20 | 0.9019 | 0.8516 | 0.7885 | 0.8666 |
| 30 | 0.8641 | 0.8366 | 0.7563 | 0.8456 |
| 40 | 0.8464 | 0.8345 | 0.7426 | 0.8345 |
| 50 | 0.8332 | 0.8553 | 0.7468 | 0.8400 |
| 60 | 0.8626 | 0.8127 | 0.7377 | 0.8312 |
| 70 | 0.8399 | 0.8303 | 0.7326 | 0.8303 |
| 80 | 0.8310 | 0.8395 | 0.7323 | 0.8310 |
| 90 | 0.8197 | 0.8829 | 0.7464 | 0.8414 |

### 5.3. Comparison of the Proposed Methods

Table 6 shows a comparison between the three approaches developed in this work. In order to compare the performance, we selected the score $F_1$, which balances both precision and recall metrics. It can be observed that better results are obtained when the

proportion is 80%, as mentioned previously. The method with the highest $F_1$ corresponds to the formant (SVM) with 95.5%.

According to what is shown in Table 6, for the case in which the majority of the audios used for validation (16% and 32%) do not have uncertainty, the speech-to-text (SVM) method works better. While the case in which the majority of the validation data have uncertainty (64% and 80%), the method that works best is the formant (SVM). In a balanced condition (48%), the best result is delivered by speech-to-text (DNN).

**Table 6.** $F_1$ Metric comparison of both approaches showing the maximum values.

| Metric | Approach | 16% | 32% | 48% | 64% | 80% |
|---|---|---|---|---|---|---|
| | 1.0 Formant (SVM) | 11.6 | 39.6 | 68.1 | 77.4 | 95.5 |
| $F_1$ | 2.1 Speech-to-text (DNN) | 8.8 | 45.9 | 72.1 | 81.0 | 88.0 |
| | 2.2 Speech-to-text (SVM) | 49.2 | 56.5 | 67.3 | 77.3 | 90.1 |

## 6. Conclusions

In this article, we show the development of two methods for automatic uncertainty classification in verbal communications. The results demonstrate that the detection and classification of uncertainty is feasible, despite the reduced set of data available.

The first method, called formant analysis, extracts from the spectrogram the relevant frequency content of the first two formants per sample time, and determines the level of uncertainty within the audio, based on the change that these formants suffer with time. The second method, called speech-to-text, infers from the order of words used in the audio, the level of uncertainty, interpreting the sentiment of the speakers. All results, for most of the performance metrics, show that the proportion with 80%, which corresponds to 31 audios with uncertainty and 8 without uncertainty, seems to provide the best result. The results for balanced proportions of audios (48% and 64%) depict much more stable performances independently of the validation range. In terms of the limitations to the proposed method, there is some difficulty in obtaining new data which affects the performance of data-based models. And, due to the inherent nature of these types of communications, the data tend to be unbalanced. Further works should consider the collection of a larger set of data to improve the balance between the two kinds of audios.

Finally, in order to support further studies, interested readers can use the GitHub repository to test their own classifiers.

**Author Contributions:** Conceptualization, G.G. and G.F.; methodology, S.D.-C. and G.H.; software, J.A.; validation, G.G., R.R. and E.F.; formal analysis, G.F. and S.D.-C.; investigation, J.A.; resources, G.F. and S.D.-C.; data curation, G.G.; writing—original draft preparation, G.G. and J.A.; writing—review and editing, G.F. and J.A.; visualization, J.A.; supervision, R.R.; project administration, G.F.; funding acquisition, G.F., S.D.-C. and E.F. All authors have read and agreed to the published version of the manuscript.

**Funding:** This research was supported, in part, by the Chilean Research and Development Agency (ANID) under Project FONDECYT 1191188, the Ministry of Science and Innovation of Spain under Projects PID2019-108377RB-C32 and PID2022-137680OB-C32 and by Agencia Estatal de Investigación (AEI) under Project PID2022-139187OB-I00.

**Institutional Review Board Statement:** Not applicable.

**Informed Consent Statement:** Not applicable.

**Data Availability Statement:** To support further studies, a repository was created in https://github.com/IALabPUCV/CallsClassification.git GitHub with the spectrogram and tokenized words of all audios.

**Conflicts of Interest:** The authors declare no conflicts of interest.

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
