# Peer review of "Uncertainty Detection in Supervisor–Operator Audio Records of Real Electrical Network Operations"

_electronics, doi:10.3390/electronics13010141_

Round 1
Reviewer 1 Report
Comments and Suggestions for Authors
1、The training details of the proposed method should be elaborated. The detailed setting of the method is not clear, some important parameter settings should be added in the manuscript. I suggest stating this in the paper in order to allow the readers to reproduce the work.
2、Is the experimental result in the manuscript a single result or an average of multiple results? If it is an average, it is recommended to include the standard deviation to demonstrate the stability of the model's performance.
3、Many images in the manuscript are not clear enough and the author is advised to replace them all with vector images.
4、Although the proposed approach is likely to be better than some other approaches, some sort of discussion of the pros and cons of the proposed one compared to other methods would be useful. The limitations of the proposed method should also be mentioned in the manuscript.
5、There are references that have not been correctly cited.
Comments on the Quality of English LanguageMinor editing of English language required
Author Response
1. The training details of the proposed method should be elaborated. The detailed setting of the method is not clear, some important parameter settings should be added in the manuscript. I suggest stating this in the paper in order to allow the readers to reproduce the work.
Response:
Important training parameters are added to the article. The following has been added to the manuscript:
Section 3.2: “For the training a linear Kernel function was used, and a cost of penalty factor (box constraint)”.
Section 4.1.2: “For training, the following hyper-parameters were used: at the input layer, an embedding dimension of 150; at both dense layers, 60 neurons, and regularizers of 0.01; and dropout factors of 0.2. The total number of epochs was 150” .
Section 4.2: “Similarly as in Section 3.2, for the training a linear Kernel function was employed, with a cost of penalty factor”.
------------------------------------------------------------------------------------
2. Is the experimental result in the manuscript a single result or an average of multiple results? If it is an average, it is recommended to include the standard deviation to demonstrate the stability of the model's performance.
Response:
Each result in Section 5 Experimental Results, was derived from 100 runs. The mean values were included in the paper. The following sentence was added in Section 5:
“Mean values are presented.”
-----------------------------------------------------------------------------------
3. Many images in the manuscript are not clear enough and the author is advised to replace them all with vector images.
Response:
All figures were converted to vector images.
----------------------------------------------------------------------------------
4. Although the proposed approach is likely to be better than some other approaches, some sort of discussion of the pros and cons of the proposed one compared to other methods would be useful. The limitations of the proposed method should also be mentioned in the manuscript.
Response:
To the best of our knowledge, no reported works determine operational uncertainty, so it is not possible to perform a detailed comparison with other methods with similar data to the one used in the present work. This is one of the reasons why we created a GitHub account with the necessary data for the application of other methods, and could serve as a baseline for more developments.
As detected limitations of the current proposal we see the following:
- Difficulty in obtaining new data, affecting the performance of data-based models.
- Due to the inherent nature, data tends to be unbalanced.
These are added to the Conclusions.
---------------------------------------------------------------------------------------
5. There are references that have not been correctly cited.
Response:
All references were thoroughly checked and corrected if necessary.
Reviewer 2 Report
Comments and Suggestions for Authors
In the paper, the communication between an electrical company control center and factory workers on the field is chosen for analysis. The authors developed two complementary approaches by using machine learning and deep learning algorithms to assess, in an automatic way, the quality of information transmission in the voice communications. The experimental results have verified the effectiveness of the proposed method. Some suggestions are as follows:
1. The theoretical analysis in this article is insufficient. Theoretical innovation is relatively average.
2.How is the interpretability of the obtained results ensured?
3. Lack of comparison with the latest methods.
Comments on the Quality of English Language
N/A
Author Response
1. The theoretical analysis in this article is insufficient. Theoretical innovation is relatively average.
Response:
We agree that the novelty is not theoretical, but rather the contribution is in the application of machine learning in a context like the voice interaction between the control centre and the factory workers of an electrical company. To the best of our knowledge there is no work focused on this topic. For this reason we have enabled access to the data used in this work, stored in a Github account, so applications of new methods could be tested, serving as a baseline.
-----------------------------------------------------------------------------------------
2. How is the interpretability of the obtained results ensured?
Response:
The validation of data-driven models is based on a training set of data. This methodology is called cross-validation, where data is randomly selected or separated into training and validation sets. The values shown in Section 5 are average values of 100 executions of this methodologies.
---------------------------------------------------------------------------------------
3. Lack of comparison with the latest methods.
Response:
The following paragraph was added to Section 5.3 to expand and deepen the comparison:
“According to what is shown in Table 6, for the case in which the majority of the audios used for validation (16% and 32%) do not have uncertainty, the Speech-to-text (SVM) method works better. While the case in which the majority of the validation data has uncertainty (64% and 80%), the method that works best is the formant (SVM). In a balanced condition (48%) the best result is delivered by speech-to-text (DNN).“
Reviewer 3 Report
Comments and Suggestions for Authors
Recommendation: Major Revision
1. The abstract should more clearly define the "quality of verbal communication" and its measurement methods.
2. The introduction could benefit from more specific examples of communication errors in electrical networks.
3. More detailed explanations of the communication protocols and their impact on uncertainty detection are needed.
4. Expand on how the 5,000 monthly recordings are statistically analyzed for clarity and error risks.
5. Clarify the formant analysis process, particularly the identification of phonemes representing uncertainty.
6. Provide more details on machine learning metrics used and their effectiveness in uncertainty detection.
7. Elaborate on the use of deep neural networks and SVMs for word frequency analysis and uncertainty prediction.
Author Response
1. The abstract should more clearly define the "quality of verbal communication" and its measurement methods.
Response:
The Abstract is modified to better include the definition of clarity in voice communications, which in this context implies that the instructions delivered by the supervisor from the command centre, are given once with minimal confirmation from the worker in the field.
----------------------------------------------------------------------------------------
2. The introduction could benefit from more specific examples of communication errors in electrical networks. More detailed explanations of the communication protocols and their impact on uncertainty detection are needed.
Response:
The following paragraph has been added to the Introduction.
“In general, the communication goes from the command centre to the field worker. Instructions are given to the deployed personnel, and they normally acknowledge correct reception or understanding. Problems in this communication normally imply either delays in the fixing of some failed device, restoring services, or in more extreme cases, a fallen service for unnecessary extra time”.
----------------------------------------------------------------------------------------
3. Expand on how the 5,000 monthly recordings are statistically analyzed for clarity and error risks.
Response:
In this study 190 voice communications (31 labelled “with uncertainty” and 159 “without”), were a human expert did the labelling. This is a very costly and time consuming task, and every new piece of data added to the training of the models is highly valuable.
-----------------------------------------------------------------------------------------
4. Clarify the formant analysis process, particularly the identification of phonemes representing uncertainty.
Response:
Using SFFT the spectrogram of an audio is calculated. This creates a surface with time step and frequency range in the horizontal plane (see Figure 4 in the paper). From this, for each time step, the first two peaks (the first two formants) along the frequency axis are detected. Then a comparison of the frequency value of the 2 formants between adjacent spectrums (adjacent time steps) is done. The rationale is that a shift in a frequency beyond a threshold (50 Hz, empirically determined) is indicative of uncertainty. This is explained in Section 3.1.
----------------------------------------------------------------------------------------
5. Provide more details on machine learning metrics used and their effectiveness in uncertainty detection.
Response:
The metrics used on this paper are the standard ones in the machine learning context. These are explained in Section 2.
----------------------------------------------------------------------------------------
6. Elaborate on the use of deep neural networks and SVMs for word frequency analysis and uncertainty prediction.
Response:
The description of the DNN and SVM techniques are included in Sections 4.1 and 4.2. Both methods require a characteristics vector which is derived from the data, in this case being the weighted frequency of words.
Please note that the DNN does not use frequency of words for classification of texts. Instead it uses the tokenizing procedure to transform texts into numerical vectors, which are then fed to the DNN, as shown in Figure 8, Section 4.1.1.
In the case of SVM, this approach uses the method based on frequency of words, together with the NNMF technique to construct the characteristic vector, becoming the input to the support vector machine, es details in Section 4.2.
Reviewer 4 Report
Comments and Suggestions for Authors
The manuscript is well-written and presents two machine-learning-based approaches to detect uncertainties and evaluate quality of information transmission in voice communication. The following questions/clarifications may be considered:
1. The speech-to-text analysis requires the definition of a bag of words a priori, which requires qualitative human input and expertise in the specific context considered; how is this list of words determined and updated as new information presents itself?
2. In the formant analysis, a threshold noise comparison is needed filter out noise-corrupted formants. How is this threshold defined? Some statistical analyses based on large dataset would help rationalize.
3. The manuscript has delineated the effectiveness of the proposed methods in classifying uncertainties in communication. It would also be interesting to see the how this may be deployed in an actual industry setting in real-time supervisor-operator communications and evaluate its impact, considering factors such as latency of detection in addition to accuracy.
Comments on the Quality of English Language
Minor English corrections needed on sentence structures, repeated phrases, and improvement on readability (e.g., line 122, line 135-136, line 176, line 220)
Author Response
1. The speech-to-text analysis requires the definition of a bag of words a priori, which requires qualitative human input and expertise in the specific context considered; how is this list of words determined and updated as new information presents itself?
Response:
As the text is incoming, an automatic logic detects if the words are or not present in the bag of words. If not, they are added with its corresponding identification number, so the updated bag can be used to identify already visited words. This process is done automatically, with software converting voice to text.
----------------------------------------------------------------------------------------
2. In the formant analysis, a threshold noise comparison is needed filter out noise-corrupted formants. How is this threshold defined? Some statistical analyses based on large dataset would help rationalize.
Response:
The threshold was used to limit the effect of noise on the calculation of formants from the spectrograms. Due to the reduced data set, this was initially determined as trial and error. It is expected to derive a statistical base for more automatized threshold determination, by having access to a greater amount of data.
----------------------------------------------------------------------------------------
3. The manuscript has delineated the effectiveness of the proposed methods in classifying uncertainties in communication. It would also be interesting to see the how this may be deployed in an actual industry setting in real-time supervisor-operator communications and evaluate its impact, considering factors such as latency of detection in addition to accuracy.
Response:
We agree with the comment, in that this should be possible to be applied in real time. It is also interesting to analyse that there could be uncertainty associated with specific workers, leading to the improvement of training.
In the case of formants, an SFFT (short FFT) would be needed to obtain the formants in real time, which are introduced to the SVM model, already trained. In the case of bag of words, online audio-to-text conversion is required, then tokenization, and application of the NNMF. This output is then introduced into the DNN and the SVM separately.
Round 2
Reviewer 1 Report
Comments and Suggestions for Authors
There are no further questions.
Comments on the Quality of English LanguageMinor editing of English language required
Reviewer 2 Report
Comments and Suggestions for Authors
This paper can be accepted.
Comments on the Quality of English LanguageN/A
Reviewer 3 Report
Comments and Suggestions for Authors
Recommendation: Accept